# An Epitope Platform for Safe and Effective HTLV-1-Immunization: Potential Applications for mRNA and Peptide-Based Vaccines

**DOI:** 10.3390/v13081461

**Published:** 2021-07-27

**Authors:** Guglielmo Lucchese, Hamid Reza Jahantigh, Leonarda De Benedictis, Piero Lovreglio, Angela Stufano

**Affiliations:** 1Department of Neurology, Medical University of Greifswald, 17475 Greifswald, Germany; 2Interdisciplinary Department of Medicine-Section of Occupational Medicine, University of Bari, 70124 Bari, Italy; hamidreza.jahantigh@uniba.it (H.R.J.); leonarda.debenedictis@uniba.it (L.D.B.); piero.lovreglio@uniba.it (P.L.); angela.stufano@uniba.it (A.S.); 3Animal Health and Zoonosis Doctoral Program, Department of Veterinary Medicine, University of Bari, 70010 Bari, Italy

**Keywords:** HTLV-1, vaccine, peptide, adjuvant, similarity, autoimmunity

## Abstract

Human T-cell lymphotropic virus type 1 (HTLV-1) infection affects millions of individuals worldwide and can lead to severe leukemia, myelopathy/tropical spastic paraparesis, and numerous other disorders. Pursuing a safe and effective immunotherapeutic approach, we compared the viral polyprotein and the human proteome with a sliding window approach in order to identify oligopeptide sequences unique to the virus. The immunological relevance of the viral unique oligopeptides was assessed by searching them in the immune epitope database (IEDB). We found that HTLV-1 has 15 peptide stretches each consisting of uniquely viral non-human pentapeptides which are ideal candidate for a safe and effective anti-HTLV-1 vaccine. Indeed, experimentally validated HTLV-1 epitopes, as retrieved from the IEDB, contain peptide sequences also present in a vast number of human proteins, thus potentially instituting the basis for cross-reactions. We found a potential for cross-reactivity between the virus and the human proteome and described an epitope platform to be used in order to avoid it, thus obtaining effective, specific, and safe immunization. Potential advantages for mRNA and peptide-based vaccine formulations are discussed.

## 1. Introduction

Human T-cell leukemia virus-1 (HTLV-1) infection is associated with a broad spectrum of clinical manifestations such as bronchiectasis, bronchitis and bronchiolitis, dermatophytosis, community acquired pneumonia, strongyloides hyperinfection syndrome, liver, lymphoma and cervical cancer and several autoimmune/inflammatory disorders such as Sjögren’s Syndrome (SS), arthropathies, and uveitis [1]. Among the many disorders that HTLV-1 carriers may develop, two are paramount: adult T-cell leukemia/lymphoma (ATL), an aggressive and often fatal hematological malignancy that is poorly responsive to most anticancer treatments, and a chronic neurological disease, HTLV-1-associated myelopathy/tropical spastic paraparesis (HAM/TSP) [1,2]. HAM/TSP patients present a series of immunological dysfunctions and among the several existing theories related to HAM/TSP development, one of the most widely accepted suggests that an autoimmune mechanism can cause lesions by molecular mimicry. A host neuronal protein seems to be similar to Tax protein from the virus, which can cause immune cross-reaction, leading to spinal cord inflammation [3].

While in certain endemic regions the prevalence and incidence of clinical symptoms among persons infected with HTLV-1 is significantly elevated, HTLV-1 infection leads to changes in the systemic immune response even in asymptomatic patients [4]. The HTLV-1 virus infects dendritic cells, monocytes and CD4+ helper T-cells and can induce changes in the activity of regulatory CD4 T-cell molecules, affecting the homeostasis of cytokines and disrupting the balance in inflammatory and anti-inflammatory responses, leading to the loss of tolerance and the development of autoimmunity. Moreover, previous research suggested that molecular mimicry could be a possible trigger mechanism for the development of autoimmune diseases linked to HTLV-1 infection [5].

To evoke proper immune responses against HTLV-1, especially cellular immunity without undesired side effects, development of an effective and safe vaccine is essential. Synthetic peptide-based vaccines have been developed to evoke immune responses with various advantages in comparison to conventional vaccine formulations. Peptide vaccines are intended to induce cellular and humoral immunity and are chemically stable without oncogenic potential [6].

Several peptide vaccines against HTLV-1 have been proposed [7] and seem to elicit a high cellular response and a significant decrease in the proviral load as well as partial protection in immunized animals, and moreover a recently developed anti-ATL therapeutic vaccine exhibited favorable clinical outcomes [8]. However, there are still no data regarding the potential use of these vaccines in view of prophylactic administration in the general population potentially exposed to this viral disease.

With the aim of investigating possible immunotherapeutic approaches against HTLV-1, we analyzed the HTLV-1 proteome in an attempt to extract peptide sequences belonging uniquely to the virus and absent in the human proteome. The research rationale was that immunotherapeutic approaches based on peptide sequences unique to the virus might be highly immunogenic by being unknown to the immune system, thus representing the “non-self” [9]. In addition, the anti-viral immunoreactivity would also have the advantage of being exempt from cross-reactions and consequent autoimmune pathologies [10]. According to this rationale, we define here a set of HTLV-1 peptides that might represent a platform for developing safe and effective anti-HTLV-1 vaccines.

## 2. Methods

The analyzed HTLV-1 polyprotein sequence (accession NCBITaxId11926) is described at https://www.uniprot.org/proteomes/?query=11926&sort=score, accessed on 1 September 2020 [11]. The HTLV-1 polyprotein sequence was cut into pentapeptides overlapping each other by four residues, i.e., MGQIF, GQIFS, QIFSR, and IFSRS. Then each HTLV-1 pentapeptide was analyzed for exact matches in the human proteome. Matching analyses of the viral polyprotein sequence to the human proteome were conducted using the Peptide Match Program (https://research.bioinformatics.udel.edu/peptidematch/, accessed on 1 September 2020) [12]. The Immune Epitope Database and Analysis Resources (IEDB; http://www.iedb.org/, accessed on 1 September 2020) was searched for peptide sequences unique to the virus in order to assess their immunological relevance [13].

## 3. Results

### 3.1. From Sequence Similarity Analyses to Anti-HTLV-1 Vaccination: Formulating a Specific and Effective Vaccine

Figure 1 reports the similarity profile of HTLV-1 amino acid (aa) sequence to the human proteome at the 5mer level. It can be seen that the occurrences of the viral pentapeptides show a constant pattern in the human proteome, with high redundant peptide areas that regularly alternate to viral peptide areas that are rare or absent in the human proteome.

As a second step in our investigation, we searched among pentapeptides absent in the human sequences and eventually selected unique consecutive viral pentapeptides overlapping each other by four residues. The results are reported in Table 1, showing that HTLV-1 has 15 peptide stretches consisting of at least 3 pentapeptides overlapping each other by 4 residues, distributed among the viral proteins Gag polyprotein, Gag-Pro-Pol polyprotein, Envelope glycoprotein gp62, and Protein Tax-1.

Finally, the peptides described in Table 1 were analyzed for their immunologic potential as follows. HTLV-1 derived epitopes were retrieved from IEDB, and the epitopes that had been experimentally validated as immunopositive in HLA-binding, B cell, and T cell assays were analyzed for the presence of the peptide sequences unique to HTLV-1 that are presented in Table 1. Moreover, predicted binding to HLA I molecules was further investigated with NetMHC 4.0. [14]. Heptapeptides were extended by 1 aminoacid (aa) residue, as naturally occurring in the viral proteome, in the N-terminus for binding prediction. Experimentally validated and predicted binding to HLA class I molecules information is presented in Table 1.

Data from B cell and T cell assays as retrieved from the IEDB are reported in Table 2. It was found that the majority of the unique HTLV-1 peptides are present in immunoreactive epitopes with the majority of them allocated in HTLV-1 envelope glycoprotein, in accordance with the predominant role of the immune response against HTLV-1 envelope glycoprotein [15].

### 3.2. From Sequence Similarity Analyses to Autoantibodies Generation: Potential Contribution to the HTLV-1 Pathological Burden

The data reported above not only open new scenarios for effective and specific anti-HTLV-1 vaccine formulations, but can also provide a potential mechanistic link between HTLV-1 infection and related disorders including lymphomas and leukemias, respiratory disorders, fibromyalgia, rheumatoid arthritis, arthritis, tuberculosis, kidney and bladder infections, dermatophytosis, community acquired pneumonia, strongyloides hyperinfection syndrome, liver cancer, lymphoma, cervical cancer, and neurological disorders [1,2].

Indeed, analyses of the experimentally validated epitopes described in Table 2 for peptide sharing with human proteins reveals that, coherently with Figure 1, the HTLV-1-derived epitopes are mostly composed of peptide sequences present in human proteins that, when altered, mutated, deficient or not functioning, may associate with numerous diseases related to HTLV-1 infection, thus supporting the possibility that cross-reactivity between the virus and human proteins can induce pathogenic autoantibodies and the HTLV-1 diseasome.

Generation of autoantibodies following peptide sharing-induced cross-reaction would also explain the multitude of disorders that, to a greater or lesser extent, characterize HTLV-1 infection. Indeed, analyses of the peptide overlap between the viral epitopes and the human proteome show that, on the whole, 64 epitope-derived hexapeptides repeatedly occur among 143 human proteins (Table 3), thus providing an ample platform for cross-reactions and, consequently, outlining a wide post-infection severe pathologic sequela.

Reasons of space hamper a protein-by-protein discussion and here we focus on some potentially biologically relevant target proteins (identified by their UniProt name; peptides shared with HTLV-1 are given in parentheses) and the related pathologies that could arise in case of cross-reactivity. Examples among the many are:A-kinase anchor protein 1, mitochondrial precursor (PSLALP) has a pivotal role in mitochondrial physiology, and its degradation leads to an increase in reactive oxygen species production, mitochondrial dysfunction, and ultimately cell death [16].A-kinase anchor protein 7 isoform gamma (HLTLPF) modulates L-type Ca2+ channels [17]. Alterations of L-type Ca2+ channels may affect cardiac contraction [18], associate with diabetes [19] and can cause autism [20] and relate to Alzheimer’s disease, Parkinson’s disease, Huntington’s disease, neuropsychiatric diseases, and other CNS disorders [21].Ataxin-2-like protein (QAIVSS) is involved in spinocerebellar ataxia type 2 [22], cutaneous T-cell lymphomas [23], is directly relevant to allergic disease [24], and its absence triggers mid-gestational embryonic lethality, affecting female animals more strongly [25].B-cell CLL/lymphoma 6 member B protein and B-cell CLL/lymphoma 9-like protein, both of which share the peptide PPTAPP with HTLV-1. When altered, these two proteins relate to tumorigenesis [26].Cadherin-2 (APQVLP) may be associated with reduction in bone mineral density or vertebral fracture prevalence in survivors of childhood acute lymphoblastic leukemia [27].Constitutive coactivator of PPAR-gamma-like protein 2 (QLPPTA) deletion is associated with autism [28].Dedicator of cytokinesis protein 9 (PPLLPH, PLLPHS, SLALPA) contributes to both risk and increased illness severity in bipolar disorder [29].DNA methyltransferase 1-associated protein 1 (APPLLP) is an essential regulator of activity and function of Ataxia Telangiectasia Mutated (ATM) kinase [30]. Alterations of the function and activity of ATM cause severe disability, poor coordination and telangiectasia, i.e., small dilated blood vessels [31].DNA polymerase nu (PPPGPC) plays a role in translesion DNA synthesis during interstrand cross-link repair in human cells [32].E3 ubiquitin-protein ligase HECTD1 (APGYDP) is an important factor in promoting base excision repair (BER), which is the major cellular DNA repair pathway that recognizes and excises damaged DNA bases to help maintain genome stability [33].Fertilization-influencing membrane protein (PSQLPP) plays a role in sperm–oocyte fusion during fertilization [34].FLYWCH-type zinc finger-containing protein 1 (PSLALP) has a possible tumor suppressive role in preventing colorectal cancer metastasis [35].Heterogeneous nuclear ribonucleoprotein C-like 1 (DLQAIK, LQAIKQ, QAKQE), alterations of which are found in sporadic and suspected Lynch syndrome endometrial cancer [36].Histone acetyltransferases KAT2A and KAT2B (DGRVIG), when altered, are associated with cardiovascular pathology [37].Islet cell autoantigen 1 (MKDLQA) Islet autoantibodies are typically associated with type 1 diabetes, but have been found in patients diagnosed with type 2 diabetes in whom they are associated with lower adiposity [38].La-related protein 1B (AIKQEV) is dysregulated in hepatocellular carcinoma [39].Mitochondrial dynamics protein MID49 (QPRPPP) is related to myopathies [40].Mucin-5AC precursor (SLSPVP, LSPVPT) is a major macromolecular component that determines the rheological properties of mucus; otherwise abnormal not-flowing mucus results in defective lung protection and leads to infection and inflammation [41].Neurofascin precursor (TGAVSS), which is a cell adhesion, ankyrin-binding protein, may be involved in neurite extension, axonal guidance, synaptogenesis, myelination, and neuron–glial cell interactions. Anti-neurofascin antibodies relate to optic, trigeminal, and facial neuropathy [42].Palmitoyltransferase ZDHHC1 (LALPAP) is involved in innate viral immune response [43].Protein piccolo (PPNHRP) is required for the development and function of neuronal networks formed between the brainstem and cerebellum. Alterations of protein piccolo may result in impaired motor coordination, cerebellar network dysfunction and pontocerebellar hypoplasia [44].Serine/threonine-protein kinase WNK1 (APQVLP) can cause hereditary sensory and autonomic neuropathy [45].Suppressor of tumorigenicity 7 protein (SLILPP) deficiency promotes laryngeal squamous cell carcinoma [46].Finally, to conclude this survey of pathologies that may be associated with HTLV-1 infection following cross-reactivity and autoantibody generation, we highlight the HTLV-1 epitope-derived VNFTQE peptide that is present in seventeen zinc finger proteins (ZFP) (numbered in Table 3 as: 57 homolog, 69, 101, 124, 136, 334, 439, 440, 442, 563, 669, 700, 709, 763, 823, 844, and 878). Zinc finger proteins comprise transcription factor families that are known for their ability to bind Zn2+ and are associated with numerous disorders. In the present context, the following are noteworthy: ZFP823 family, when mutated or dysregulated, is involved in acute leukemias [47]. Altered/dysregulated ZFPs may also associate with neurodevelopmental disorders such as intellectual disability, autistic features, psychiatric problems, and motor dysfunction [48]. Moreover, zinc finger proteins are implicated in the development and progression of several types of cancer [49]. Simply put, antibodies against the epitope-derived peptide VNFTQE alone might be sufficient to determine the wide range of pathologies that, in more or less severe forms, afflict HTLV-1-infected patients.

## 4. Discussion

Vaccines are a crucial tool for fighting viral infections and their profile of efficacy and safety is of paramount importance. Minimizing the risk of collateral autoimmune cross-reactivity might be advantageous for further optimizing the efficacy–safety tradeoff, as recently shown by the SARS-CoV-2 global vaccination campaign [50,51].

Hence, in order to design a safe and effective HTLV-1 vaccine, we applied the scientific rationale according to which vaccines based on peptides unique to the infectious pathogens and absent in the human host might avoid cross-reactivity phenomena and offer high potential for immunogenicity and self-adjuvanticity [52,53]. We found a peptide platform consisting of 15 viral sequences (overall length: 104 aa) that are unique to the HTLV-1 polyprotein when compared to the *Homo sapiens* proteome. These sequences show an immunological potential by being embedded in epitopes that were experimentally validated in the human host, as shown in Table 2. Of note, the vast majority of such epitopes are characterized by B cell immunoreactivity and only to a minor degree by cell-mediated immune response, thus suggesting mainly humoral protective immunity. Humoral immunity is not restricted to extracellular antigens, as it can also neutralize viruses within infected cells and bind intracytoplasmic antigens [54,55]. Antibodies induced by our peptide platform might therefore be able to interact with viral proteins within HTLV-1-infected cells [56].

### 4.1. Translational Potential and Advantages for mRNA Vaccines

Messenger ribonucleic acid (mRNA) vaccines are the results of decades of studies on different mRNA intracellular delivery technologies and have come under the spotlight as one of the main tools of the global vaccination campaign against the severe acute respiratory syndrome coronavirus 2 (SARS-CoV-2) [57]. Delivery is crucial to this formulation technology because of the inherent instability of mRNA and lipid nanoparticle carriers have been adopted in order to strike the best compromise between stability and effective intracellular release [58]. Nevertheless, the real-world logistics of the mRNA vaccines during the pandemic has proven to be challenging, requiring, for instance, strict cold chain integrity [59], and product integrity issues have appeared to affect large-scale production [60]. Of relevance, the larger size of mRNA as compared, for example, to small interfering RNA or other smaller molecules complicates efficient encapsulation in the nanoparticles [57,61]. Moreover, mRNA size inversely correlates with translational rate [62,63]. The epitope platform we presented here consists of 15 epitopes for a total of 104 residues that can be extended by spacer links to ~400 aa to optimize translation [64]. Such length would correspond to a third of the size of the SARS-CoV-2 spike protein encoded by the only currently available mRNA vaccines and, by comparison, might lead to more efficient encapsulation and, after delivery, translation of the mRNA, eventually resulting in optimal translation, antigen production, and immunization.

### 4.2. Translational Potential and Advantages for Peptide-Based Vaccines

The epitope platform presented above can also be adopted in non-genetic peptide-based vaccine formulations, where the peptides are directly administered, possibly conjugated to a carrier [65]. Such formulations are easy to mass-produce, are chemically stable, and do not pose any risk of infectious or oncogenic potential. Our specific uniquely viral epitope platform additionally minimizes the risk of autoimmunity with potential high self-adjuvanticity [53]. Crucially, we previously identified, with the same approach adopted in the present work, potential epitopes of SARS-CoV-2 [52] and a number of them were subsequently experimentally confirmed to be safe and effective in eliciting immunity against the virus with a peptide-based vaccine [66,67].

## 5. Conclusions

We presented here an epitope platform for effective HTLV-1 vaccination that minimizes the risks intrinsic to whole viral antigens, thus potentially reducing post-vaccination adverse events, cross-reactivity with human antigens and possible autoimmune pathologies [68].

The short epitopes we identified are optimally suited for mRNA and nanoparticle vaccine technology, offering remarkable advantage in terms of production quality and stability of the final products, thus easing the currently challenging logistic aspects of storage and transportation. Moreover, the same epitope platform is a flexible resource that can also be used in peptide-based vaccines at lower cost, facilitating production and diffusion, for instance in developing countries.

In sum, we presented an epitope set that offers remarkable real-world logistic advantages for a large-scale safe and effective prophylactic vaccination campaign against HTLV-1 that might be of cogent importance considering that no specific therapy exists.

## Figures and Tables

**Figure 1 viruses-13-01461-f001:**
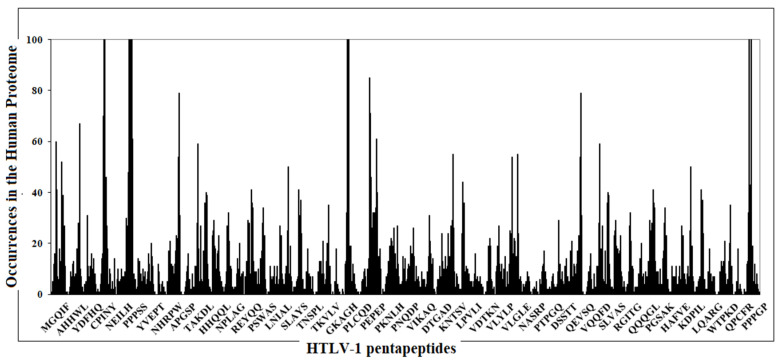
Matching profile of the HTLV-1 aa sequence to the human proteome at the pentapeptide level.

**Table 1 viruses-13-01461-t001:** Peptides that are ≥7mer long and consist of overlapping minimal 5-mer immune determinants unique to HTLV-1 and absent in the human proteome.

HTLV-1 Protein	AaPosition	Peptide ^1^	MHC Binding ^2^
Gag polyprotein	132–138	VMHPHGA	**HLA-B*07:02**
Gag polyprotein	165–171	PQFMQTI	*HLA-B*07:02*
Gag polyprotein	331–337	ACQTWTP	n.a.
Gag polyprotein	364–370	GHWSRDC	n.a.
Gag-Pro-Pol polyprotein	915–921	SKEQWPL	n.a.
Gag-Pro-Pol polyprotein	1086–1092	RSWRCLN	n.a.
Gag-Pro-Pol polyprotein	1351–1358	IALWTINH	n.a.
Gag-Pro-Pol polyprotein	1366–1374	HKTRWQLHH	n.a.
Gag-Pro-Pol polyprotein	1390–1398	KQTHWYYFK	*HLA-A*03:01* *HLA-B*27:05*
Gag-Pro-Pol polyprotein	1429–1435	SAQWIPW	**HLA-A*24:02** **HLA-A*02:01** **HLA-B*35:01** *HLA-B*58:01*
Protein Tax-1	49–56	CPEHQITW	n.a.
Envelope glycoprotein gp62	116–122	GCQSWTC	n.a.
Envelope glycoprotein gp62	131–137	PYWKFQH	n.a.
Envelope glycoprotein gp62	170–176	YDPIWFL	n.a.
Envelope glycoprotein gp62	272–278	NWTHCFD	n.a.

^1^ Identified as described under Methods and in text. ^2^ Detailed methods and discussion and in text. n.a. = data not available. Bold indicates experimentally validated biding to the allele as reported in the IEDB; italics indicates predicted binding to the HLA-allele.

**Table 2 viruses-13-01461-t002:** Distribution among experimentally validated epitopes that are immunoreactive in the human host of peptides that are ≥7mer long, consist of overlapping minimal 5-mer immune determinants unique to HTLV-1, and are absent in the human proteome. ^1^ (www.iedb.org, accessed on 1 September 2020). ^2^ HTLV-1 peptides unique to HTLV-1 and absent in the human proteome are given in capital letters.

IEDB ID ^1^	EPITOPE SEQUENCE ^2^	VIRAL PROTEIN	ASSAY
7612	dapgYDPIWFLntepsqlpptappllphsnldhile	Envelope glycoprotein	B cell
17248	fNWTHCFDpqiqaivsspchnslilppfslspvpt	Envelope glycoprotein	B cell
18895	GCQSWTCpytgavssPYWKFQHdvn	Envelope glycoprotein	B cell
20199	GHWSRDCtqprpppgpcplcqdp	Pr gag-pro-pol.1	B cell
34858	lalpaphltlpfNWTHCFDpqiq	Envelope glycoprotein	B cell
38482	lpfNWTHCFDpq	Envelope glycoprotein	B cell
47047	pcslkcpylGCQSWTCpytgavs	Envelope glycoprotein	B cell
49915	pVMHPHGAppnhrpwqmkdlqaikqevsqa	Pr gag-pro-pol.1	B cell
59328	sllvdapgYDPIWFLntepsqlpptappllphsnldhilepsipwks	Envelope glycoprotein	B cell
61300	ssPYWKFQHdvnftqevsrln	Envelope glycoprotein	B cell
62700	sysdpcslkcpylGCQSWTCpyt	Envelope glycoprotein	B cell
65003	tlpfNWTHCFDpqiqaivs	Envelope glycoprotein	B cell
65645	tpllypslalpaphltlpfNWTHCFDpqiq	Envelope glycoprotein	B cell
78192	lalpaphltlpfNWTHCFDpqiqaivsspchnsli	Envelope glycoprotein	B cell
78715	latCPEHQITWdpidgrvig	Transcriptional activator Tax	B cell
94435	apgYDPIWFL	Envelope glycoprotein	T cell
96592	lpfNWTHCFDpqiqaivsspc	Envelope glycoprotein	T cell
97892	apqvlpVMHPHGAppn	Pr gag	B cell

**Table 3 viruses-13-01461-t003:** Peptide sharing between HTLV-1-derived epitopes and human proteins.

Peptides	Human Proteins
SLILPP	(Pyruvate dehydrogenase (acetyl-transferring)) kinase isozyme 2, mitochondrial, Cilia- and flagella-associated protein 44, F-BAR domain only protein 1, Suppressor of tumorigenicity 7 protein, Suppressor of tumorigenicity 7 protein-like
PSLALP	3-oxo-5-alpha-steroid 4-dehydrogenase 1, A-kinase anchor protein 1 mitochondrial precursor
QPRPPP	Acrosin precursor, Adipocyte enhancer-binding protein 1 precursor, Alpha-ketoglutarate-dependent dioxygenase alkB homolog 6, Copine-9, Spectrin beta chain non-erythrocytic 2, Mitochondrial dynamics protein MID49, Synapsin-3, TANK-binding kinase 1-binding protein 1, Tensin-1
PCSLKC	ADAMTS-like protein 1 precursor
HLTLPF	A-kinase anchor protein 7 isoform gamma
LALPAP	Alpha-mannosidase 2C1, Membrane-associated phosphatidylinositol transfer protein 2, Palmitoyltransferase ZDHHC1, Phosphatidylinositol 3-kinase regulatory subunit beta, Probable threonine protease PRSS50 precursor, Serine protease 56 precursor, Uncharacterized protein C11orf24 precursor
QAIVSS	Ataxin-2-like protein
PPTAPP	B-cell CLL/lymphoma 6 member B protein, B-cell CLL/lymphoma 9-like protein
SLSPVP	BCLAF1 and THRAP3 family member 3, Mucin-5AC precursor, Zinc finger and SCAN domain-containing protein 22
APQVLP	Cadherin-2 precursor
QEVSRL	Calpain-10, Caseinolytic peptidase B protein homolog precursor, Dehydrogenase/reductase SDR family member 7C precursor, Kinesin-like protein KIFC3, Leucine-rich repeat-containing protein 69, Nucleobindin-1 precursor, Transcription factor MafB
RPPPGP	Carbohydrate-responsive element-binding protein, Cyclic GMP-AMP synthase, Helicase SRCAP, Homeobox protein HMX1, Membrane protein FAM174B precursor, Monocarboxylate transporter 3, Potassium/sodium hyperpolarization-activated cyclic nucleotide-gated channel 2, Proline-rich protein Y-linked, Sprouty-related, EVH1 domain-containing protein 3
APGYDP	Carboxypeptidase N catalytic chain precursor
TQPRPP	CCR4-NOT transcription complex subunit 3
KDLQAI, DLQAIK	Cilia- and flagella-associated protein 44
VLPVMH	Coiled-coil domain-containing protein 77
QLPPTA	Constitutive coactivator of PPAR-gamma-like protein 2
AIVSSP	Cullin-9
CTQPRP	Cyclin-G-associated kinase
PPLLPH	Dedicator of cytokinesis protein 9
PLLPHS	Dedicator of cytokinesis protein 9, LIM/homeobox protein Lhx8, Probable palmitoyltransferase ZDHHC11B
SLALPA	Dedicator of cytokinesis protein 9, Transmembrane protein 130 precursor
APPLLP	DNA methyltransferase 1-associated protein 1, Protein AF-17, Synaptojanin-2, Transcription factor HES-4
PPPGPC	DNA polymerase nu, Protein FAM214B, Tastin
NTEPSQ	DNA topoisomerase 2-binding protein 1
IDGRVI	DnaJ homolog subfamily B member 5
PLCQDP	Dynein heavy chain 14, axonemal
QEVSQA	Dynein heavy chain 3, axonemal
APGYDP	E3 ubiquitin-protein ligase HECTD1
LPPFSL	E3 ubiquitin-protein ligase ZNF598
PYTGAV	Endoplasmic reticulum resident protein 29 precursor
PSQLPP	Fertilization-influencing membrane protein, Mediator of RNA polymerase II transcription subunit 15, Mitochondrial import inner membrane translocase subunit Tim17-B, Protein KRBA1, Zinc finger and BTB domain-containing protein 32
PSLALP	FLYWCH-type zinc finger-containing protein 1
TAPPLL	Forkhead box protein Q1
PPTAPP	Forkhead box protein Q1, Immediate early response gene 2 protein
LQAIKQ, QAKQE	Heterogeneous nuclear ribonucleoprotein C-like 1
DLQAIK	Heterogeneous nuclear ribonucleoprotein C-like 1, Heterogeneous nuclear ribonucleoproteins C1/C2
DGRVIG	Histone acetyltransferase KAT2A, Histone acetyltransferase KAT2B
LYPSLA	Host cell factor 2
YPSLAL	Integrin beta-5 precursor
MKDLQA	Islet cell autoantigen 1
GAVSSP	Kallikrein-9 precursor
PRPPPG	Carbohydrate-responsive element-binding protein, Cyclic GMP-AMP synthase, Homeobox protein HMX1, Kelch domain-containing protein 7A, Microtubule organization protein AKNA, Proline-rich protein 33, RING finger protein 225, Splicing factor 3B subunit 4, Tau-tubulin kinase 2, Trinucleotide repeat-containing gene 6B protein, Proline-rich protein, Y-linked
NLDHIL	Kinocilin
AIKQEV	La-related protein 1B
DPQIQA	Mediator of RNA polymerase II transcription subunit 17
EPSQLP	Membrane primary amine oxidase
LSPVPT	Mucin-5AC precursor
TQPRPP	N-acetyl-beta-glucosaminyl-glycoprotein 4-beta-N-acetylgalactosaminyltransferase 1
TGAVSS	Neurofascin precursor
LLYPSL	Olfactory receptor 6P1
PLLYPS	Olfactory receptor 6P1, Trace amine-associated receptor 5
ALPAPH	PR domain zinc finger protein 12
PPGPCP	Proline-rich protein 13
AIKQEV	Protein C-ets-1
SQLPPT	Protein KRBA1
PPNHRP	Protein piccolo
PPTAPP	Protein sidekick-2 precursor, Stanniocalcin-2 precursor
PFSLSP	Protein-glucosylgalactosylhydroxylysine glucosidase
LVDAPG	Receptor-type tyrosine-protein kinase FLT3 precursor
APQVLP	Regulating synaptic membrane exocytosis protein 1, Serine/threonine-protein kinase WNK1
GAVSSP	RELT-like protein 1 precursor
LPPTAP	SH3 and multiple ankyrin repeat domains protein 1
TGAVSS	Sorbin and SH3 domain-containing protein 1
TQPRPP	Spectrin beta chain, non-erythrocytic 2
KFQHDV	Sterile alpha motif domain-containing protein 9-like
LATCPE	Testis-expressed protein 38
KDLQAI	Thrombospondin-1 precursor
ILEPSI	WW domain-binding protein 4
CPLCQD	Zinc finger homeobox proteins 3 and 4
LEPSIP	Zinc finger homeobox protein 4
VNFTQE	Zinc finger protein 57 homolog, Zinc finger proteins 69, 101, 124, 136, 334, 439, 440, 442, 563, 669, 700, 709, 763, 823, 844, 878
DVNFTQ	Zinc finger protein 136

## Data Availability

All data are either contained in the article or available at the links provided in the article.

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
