# Peer review of "An Epitope Platform for Safe and Effective HTLV-1-Immunization: Potential Applications for mRNA and Peptide-Based Vaccines"

_viruses, 2021, doi:10.3390/v13081461_

Round 1

Reviewer 1 Report

Thank you for submitting this descriptive paper. In my opinion it can be accepted and does not need further amendments. Thank you

Author Response

We thank the Reviewer for the positive evaluation of our work.

Reviewer 2 Report

Lucchese et al. in their research manuscript entitled ‘An epitope platform for safe and effective HTLV-1-immunization: potential applications for mRNA and peptide-based vaccines’ used a sliding window approach to identify oligopeptide sequences unique to HTLV-1. Unique viral peptides were then analyzed for immunological relevance using the immune epitope database. The authors found several peptide stretches within HTLV-1 which could serve as candidates for a safe HTLV-1 vaccine. Importantly, these peptide sequences have already been experimentally validated in the human host from previous studies. The manuscript is well written and of high interest to both the HTLV-1 field and the wider scientific community. A few minor comments are provided below for the authors’ consideration:

  1. Several of the peptides were identified within the viral envelope. This logically makes sense as it is expressed on the outside of the virion. Since so many of the peptides are expressed within the same viral mRNA, this might argue that simply using the Env mRNA vs. testing several mRNA peptide-based vaccines is a better approach. This gives the advantage of providing several peptides of interest at once and could potentially provide a broader immune response. By simply using one peptide, the virus has the ability to mutate and evade immune detection. Also, what is the conservation of HTLV-1 clinical isolates within these viral peptides?
  2. Would your proposed mRNA vaccine be used as a therapeutic or prophylactic vaccine? This is an important point to address since certain disease states of HTLV-1 (ATL specifically) do not express sense genes in a majority of cases.
  3. Would it be better to elicit a B cell response or T cell response in the context of HTLV-1 (asymptomatic infection, disease states)?  The virus integrates into the host genome and thus has the ability to persist in latent reservoirs (similar to HIV-1), thus adding a layer of complexity for treatment.

Author Response

We thank the Reviewer for finding merit in our work and for the extremely relevant and useful comments.

  1. Thank you for highlighting this crucial issue and allowing us to address it. We definitely agree that using the whole pool of peptides we identified instead of only one or a few of them would guarantee higher efficacy and broader protections against various viral isolates, given that for some of the epitopes we identified experimental evidence of high conservation (Mizuguchi M, Takahashi Y, Tanaka R, Fukushima T, Tanaka Y. Conservation of a Neutralization Epitope of Human T-cell Leukemia Virus Type 1 (HTLV-1) among Currently Endemic Clinical Isolates in Okinawa, Japan. Pathogens. 2020 Jan 27;9(2):82. doi: 10.3390/pathogens9020082. PMID: 32012672; PMCID: PMC7168584.) is already available. This is the reason why we proposed to encode all of the presented peptides in a single mRNA sequence. On the other hand, adopting the mRNA sequence of the envelope protein would not only exclude many of the peptides we identified but also include aa sequences potentially common to the human proteome, thus introducing the risk of cross-reactivity.

  2. The Reviewer indeed addresses another very relevant point. Given the high safety profile and the absence of potential cross-reactivity, the peptide based vaccine we propose would be highly advantageous and effective in eradicating HTLV-1 without relevant risk and thus perfectly and ethically suited for a prophylactic strategy.

  3. Once more the Reviewer focused on an important aspect of our work, thank you. The majority of our linear epitopes are likely to elicit prevalently humoral immunity, as we state in the manuscript. Humoral immunity is not restricted to extracellular antigens, it can neutralize viruses within infected cells and bind intracytoplasmic antigens (Mallery DL, McEwan WA, Bidgood SR, Towers GJ, Johnson CM, James LC. Antibodies mediate intracellular immunity through tripartite motif-containing 21 (TRIM21). Proc Natl Acad Sci U S A. 2010 Nov 16;107(46):19985-90. doi: 10.1073/pnas.1014074107. Epub 2010 Nov 2. PMID: 21045130; PMCID: PMC2993423.; Greenlee JE, Clawson SA, Hill KE, Wood B, Clardy SL, Tsunoda I, Carlson NG. Anti-Yo antibody uptake and interaction with its intracellular target antigen causes Purkinje cell death in rat cerebellar slice cultures: a possible mechanism for paraneoplastic cerebellar degeneration in humans with gynecological or breast cancers. PLoS One. 2015 Apr 17;10(4):e0123446. doi: 10.1371/journal.pone.0123446. PMID: 25885452; PMCID: PMC4401511.). Antibodies induced by our peptide platform might therefore be able to interact with viral proteins within HTLV-I infected cells (Enose-Akahata Y, Abrams A, Massoud R, Bialuk I, Johnson KR, Green PL, Maloney EM, Jacobson S. Humoral immune response to HTLV-1 basic leucine zipper factor (HBZ) in HTLV-1-infected individuals. Retrovirology. 2013 Feb 13;10:19. doi: 10.1186/1742-4690-10-19. PMID: 23405908; PMCID: PMC3584941.)